# A Personalized Multi-Turn Generation-Based Chatbot with Various-Persona-Distribution Data

Shihao Zhu [1], Tinghuai Ma [1,*], Huan Rong [2] and Najla Al-Nabhan [3]

1 School of Software, Nanjing University of Information Science & Technology, Nanjing 210044, China
2 School of Artificial Intelligence, Nanjing University of Information Science & Technology, Nanjing 210044, China
3 Department of Computer Science, King Saud University, Riyadh 11362, Saudi Arabia
* Correspondence: thma@nuist.edu.cn

**Abstract:** Existing persona-based dialogue generation models focus on the semantic consistency between personas and responses. However, various influential factors can cause persona inconsistency, such as the speaking style in the context. Existing models perform inflexibly in speaking styles on various-persona-distribution datasets, resulting in persona style inconsistency. In this work, we propose a dialogue generation model with persona selection classifier to solve the complex inconsistency problem. The model generates responses in two steps: original response generation and rewriting responses. For training, we employ two auxiliary tasks: (1) a persona selection task to fuse the adapted persona into the original responses; (2) consistency inference to remove inconsistent persona information in the final responses. In our model, the adapted personas are predicted by an NLI-based classifier. We evaluate our model on the persona dialogue dataset with different persona distributions, i.e., the persona-dense PersonaChat dataset and the persona-spare PersonalDialog dataset. The experimental results show that our model outperforms strong models in response quality, persona consistency, and persona distribution consistency.

**Keywords:** open-domain dialogue system; persona consistency learning; speaking style learning; generation-based chatbot





## 1. Introduction

Building a stable human-like dialogue system has been an important topic in artificial intelligence. Some of them are now widely used in daily life, such as chit-chat agents [1], question-and-answer systems [2], etc. Dialogue systems can be divided into two categories: open-domain and task-oriented. Open-domain dialogue models, called chatbots, converse without definite goals on infinite topics. This has attracted an increasing number of researchers, and various open-domain dialogue models have been proposed. Among them are retrieval-based methods and generation-based methods. Unlike retrieval models, dialogue generation models can generate responses that do not exist in the training corpus. Therefore the generated dialogues may be inconsistent with the context and the given information. For the consistency of the dialogue system, Huang et al. [3] classified existing research into three categories: (1) persona consistency modeling, including implicit and explicit methods; (2) stylistic response generation; and (3) contextual consistency. As shown in Figure 1, persona consistency focuses on whether the persona information in the response is semantically consistent with the given persona. Stylistic consistency is concerned with the consistency between the style of the response and the target style. Contextual consistency is concerned with whether there is an apparent contradiction between sentences, i.e., natural language inference (NLI). These are three crucial factors for constructing a consistent human-like dialogue agent. Most existing persona models focus on persona consistency modeling, aiming to avoid generating responses that conflict with the given persona. Some multi-turn dialogue persona models consider contextual consistency to generate coherent

and consistent dialogue; however, fewer studies have focused on the effect of speaking style consistency in persona models.

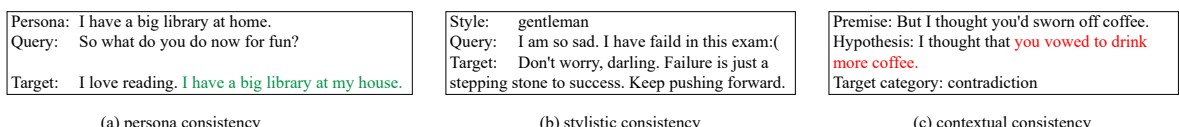

|  |
|---|
| Persona: I have a big library at home. |
| Query: So what do you do now for fun? |
| Target: I love reading. I have a big library at my house. |

(a) persona consistency

|  |
|---|
| Style: gentleman |
| Query: I am so sad. I have faild in this exam:( |
| Target: Don't worry, darling. Failure is just a stepping stone to success. Keep pushing forward. |

(b) stylistic consistency

|  |
|---|
| Premise: But I thought you'd sworn off coffee. |
| Hypothesis: I thought that you vowed to drink more coffee. |
| Target category: contradiction |

(c) contextual consistency

**Figure 1.** The examples of persona, stylistic, and contextual consistency. In example (**a**), the response contains information that is consistent with persona. For example (**b**), the response presents a gentlemanly style. The purpose of example (**c**) is to determine the consistency between premises and hypotheses.

Many recent studies on persona modeling show good persona metrics, such as Per-CVAE [4], DHAP [5], PS-Transformer [6], etc. These methods add additional persona information to obtain responses, as shown in Figure 1a, and we refer to these sentences containing persona information as persona-relevant responses. However, these models focus on the semantic consistency of persona information with content and ignore stylistic consistency. We consider that the agent chooses inappropriate personas in the current conversation or even uses persona information forcibly, causing the persona inconsistency in speaking style, i.e., a persona distribution inconsistency. In diverse contexts, people commonly adopt different speaking styles. According to Tsay-Vogel et al.'s study [7], people are more willing to self-disclose in a private chat context and less in a social media context. In fact, less than 10% of posts on Twitter are related to user persona information [8]. However, our experiments find that existing persona-based models cannot adopt different speaking styles naturally, for example the EDU$_{BOB}$ model [9]. On a dense dataset, such as PersonaChat [10], our experiment determines that 46.9% of the responses in this dataset are persona-relevant, and responses related to persona make up 20% of the responses generated by the EDU$_{BOB}$ model. On a sparse dataset, such as PersonalDialog [11], the proportion of persona-relevant responses on the dataset and EDU$_{BOB}$ was 1.2% and 3.2%.

Inspired by this, we try to alleviate persona inconsistency from the perspective of persona distribution. Our model proceeds in two steps: (1) the model learns the persona distribution from the dataset to select the adapted persona, and (2) the model fuses the selected persona. For persona selection, most of the existing models incorporate the whole persona representation into the hidden state via the attention mechanism. However, the model has not learned the strategy of persona selection, as shown in Figure 2. Instead, we try to find the most adapted persona and use it to generate style-consistent responses. This paper aims to construct an agent that can use the adapted persona information and generate persona-consistent responses with a given dataset.

The task aims to let the agent generate high-quality responses in three stages: 1. select suitable targets from given personas; 2. generate persona-relevant responses based on the selected personas; 3. ensure that the generated responses are consistent with the personas. In the first stage, we try to train a model to select an adapted persona based on historical context, so we need responses with persona labels as training data. In existing datasets, personas are contained in responses implicitly. To solve the problem, we can use the NLI model to infer the persona labels as a dataset to train it to select an adapted persona. In the second stage, we can use various large-scale persona datasets to train the dialogue generation model. In the third stage, since the initial task of the NLI model is to determine the consistency of utterance pairs, we also use the NLI dataset to train the rewritten model to obtain the final responses. Finally, we build a bidirectional decoder $D_2$ to fuse the original response into the final response $R_2$, while fusing the persona information to revise the inconsistent information in the corners of the final responses.

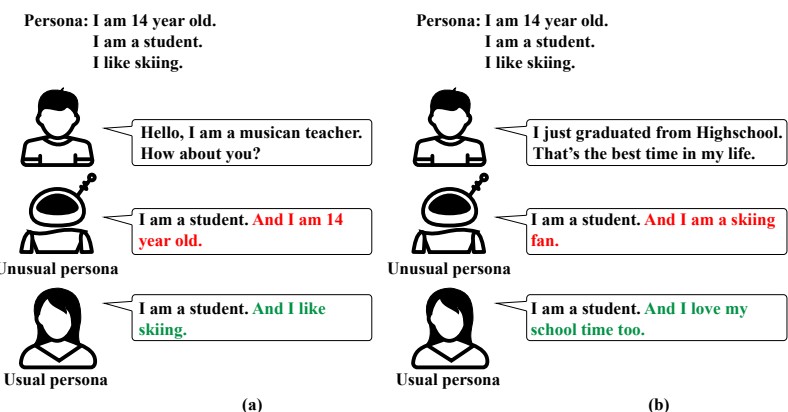

**Figure 2.** Examples of conversation using different persona. (**a**) Humans usually do not show private information in the beginning. (**b**) To show persona, agent selects the persona "I like skiing", although it would be better not to select the other given persona.

Contributions in this work are three-fold:

- We propose a BERT-based generation framework, which considers the distribution of persona in the dataset to generate persona-consistent responses.
- We designed a persona selection mechanism that explicitly selects a persona using an NLI model and implicitly fuses it into the responses. This allows the agent to exhibit different user speaking styles.
- We use the NLI model to annotate the persona of the responses, solving the problem of existing datasets without persona labels. PersonaChat dataset and PersonalDialog dataset are extended and manually evaluated with 88.5% accuracy.

## 2. Related Work

### 2.1. Persona-Based Models

In early studies on dialog systems, persona information is introduced to construct personalized agents [12]. Recent persona-based dialogue generation models are based on the data-driven approach [1,11,13], i.e., learning persona-relevant features from the large-scale persona dialogue datasets. Among these methods, persona information can be classified into three types: implicit persona embeddings [12,14], explicit profiles [13], and personal descriptions [15]. In these methods, the agent adds persona information to generate persona-related responses, a side effect of which is that the quality of the conversation is significantly reduced. With the development of large-scale pre-training models, various pre-training methods are proposed to enhance dialogue quality, such as T5 [16] and BART [17]. Zheng et al. [18] proposed a GPT-based model with the attention routing mechanism that can adjust the involvement of persona information in the generation step. Song et al. [19] designed a BERT-based model with dual decoders that can improve the quality of generated responses. Referring to their pre-training approach, we also employ Transformer as our base model and initialize it with BERT to improve conversational fluency. In addition, we refer to attention routing to fuse personas and context representations into hidden states to control the weight of persona information in responses.

Although the persona information is incorporated into the generated responses, the persona-relevant responses may contradict the given persona. Because several words in response may lead to opposite persona information, such as "I am a student" and "I am not a student." Many persona consistency models have been proposed to resolve semantic inconsistency, such as three-stage response generation [20], which rewrites the original generated dialogue and removes inconsistent information through a double decoder structure, and the personalized hybrid matching network [21], which extracts persona information from user history conversations to improve match probability. Mesgar et al. [22] proposed a reinforcement learning method with an efficient reward that can captures the semantic

consistency between responses and personas. In additon, there are some model-agnostic methods. For example, Cao et al. [23] improved the performance of strong dialogue models by curriculum training, where the language model is trained with augmented and original data. Moreover, some methods introduce persona information as external knowledge. Fu et al. [24] introduced persona to select personal knowledge in knowledge-grounded conversation, which is a closed loop. These approaches alleviate persona semantic inconsistencies, but they also have some problems. As mentioned in Section 1, these models do not take speaking style consistency into account. Our model not only considers persona semantic consistency but also focuses on the effect of persona distribution on consistency so that it can show better performance in datasets with different distributions.

### 2.2. Natural Language Inference

The natural language inference (NLI) task is to study whether a hypothesis can be inferred from a premise. The relationship between the premise and the hypothesis is usually classified into three categories: entailment, neural, and contradiction. In early studies, most deep-learning-based NLI approaches rely on the SNLI large-scale corpus [25]. Welleck et al. [26] tried to solve the dialogue consistency problem with the NLI method and constructed a dialogue consistency dataset DNLI based on SNLI. Since then, there have been growing studies that try to introduce NLI datasets to improve consistency in dialogue systems. For example, Song et al. [19] used NLI datasets to improve persona consistency with the unlikelihood method. Liu et al. [6] proposed a posterior-scored transformer that can retrieve the most relevant persona from an external knowledge base. Chen et al. [27] proposed a method of learning latent entailment relations between responses as external memories, to increase the consistency and coherence with the NLI task. Inspired by these methods, we try to explore the dependency between contexts and personas with the NLI dataset, to attain two effects: (1) to infer the most relevant persona from the context and provide them to the generative model; (2) to consider the inconsistency between response and personas to generate persona-consistent dialogues.

## 3. Model

### 3.1. Task Definition

In this paper, our persona-relevant dialogue generation problem refers to using a leading agent to select an adapted persona and maintain consistency to construct a human-like dialogue system. It can be formally defined as follows: A multi-turn conversation in a dialogue context is represented as $C = \{u_1, u_2, \cdots, u_i, \cdots, u_n\}$, where $u_i$ is the $i$th turn of the conversation, $u_n$ is a query from another speaker, and $u_{<n}$ is the conversation history of the agent. An interlocutor is given multiple profiles or descriptions to represent personas in a conversation, which is denoted as $P = \{p_1, p_2, \cdots, p_m\}$. Due to the different speaking styles of the users, each utterance in the conversation may imply a given persona or not, and the adapted persona is represented as $p_{adapted}$. Then, our task is to learn a mapping from context and personas to a response $R = \{r_1, r_2, \cdots, r_m\}$, which is consistent with both given personas and context.

### 3.2. Overview

As in Figure 3, we designed a three-stage model from two terms: select adapted persona before original generation and maintain persona consistency after original generation. In the first stage, the classifier $C_{ps}$ selects the persona that is most adapted to the speaking style of the user in the context from the given personas, while the encoder $E$ encodes the context and personas. In the second stage, the decoder $D_1$ generates an original response with the context, the given personas, and the adapted persona. In the third stage, the rewriting decoder $D_2$ generates the final response with given personas and an original response.

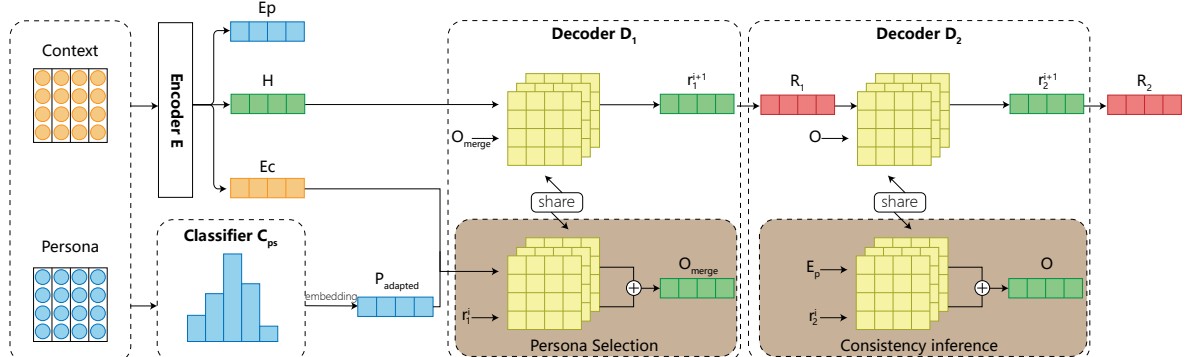

**Figure 3.** The architecture of the proposed persona-adapted and -consistent dialogue model.

### 3.3. Encoder E

The encoder *E* is a standard transformer model that encodes the word embedding as context representations. Additionally, to make it perform well, the module is initialized with BERT. The inputs of the model include context *C* and persona *P*, where the personas are unstructured on PersonaChat dataset, e.g., "gender: female" and structured on PersonalDialog dataset, e.g., "I like to listen to music." To encode into a context representation, we concatenate the context and personas into a sequence of words, denoted as

$$input = u_1, u_2, \cdots, u_m, [\text{SEP}], p_1^{(1)}, p_1^{(2)}, \cdots, p_n^{(m)} \tag{1}$$

The encoder first tokenizes the natural language input with the WordPiece method, then converts the tokenized input into a dimensionally fixed representation via the embedding layer. In usual experiments, the embedding layer performs three embeddings of the input, token, location, and type embeddings, and the sum of the three embeddings is the input representation, which is the same for our experiments too. The type embedding is 0 and 1 for persona and context, respectively. For the subsequent experiments, we also encoded the persona and context separately to obtain persona representations $E_p$ and context representations $E_c$. Next, multi-head attention converts the embedding to a sequence of hidden vectors. Multi-head attention [28] is described in detail in transformer, which is represented as MultiHead($q, k, v$), which computes the importance from query to key and value by scaled dot-product. There is a final pass through a feedforward network, denoted as FFN, which is a two-layer fully connected network with a ReLU activation function. The above modules have *N* layers in the whole encoder, hidden vectors for each layer:

$$h^{i+1} = \text{FNN}(\text{MultiHead}(h^i, h^i, h^i)) \tag{2}$$

where $h^i$ is the hidden vector of the previous layer, $h^0$ is the input representation, $h^N$ is the hidden vector of the *N*th layer and the final output of the encoder, denoted as *H*.

### 3.4. Persona-Select Classifier $C_{ps}$

The persona-select classifier is a zero-shot classification model and initialized with BART-MNLI [17] to ensure that the model is good at language inference.

The inputs in classifier $C_{ps}$ are the context and personas, as shown in Figure 4. Firstly, a special token is inserted between the context and each persona, so *n* sequences of words are obtained as follows:

$$input_{ps} = \begin{cases} \left\{ u_1, \cdots, u_m, [\text{SEP}], p_1^{(1)}, p_1^{(2)}, \cdots, p_1^{(m)} \right\}, \\ \qquad\qquad\qquad \vdots \\ \left\{ u_1, \cdots, u_m, [\text{SEP}], p_n^{(1)}, p_n^{(2)}, \cdots, p_n^{(m)} \right\} \end{cases} \tag{3}$$

Same as the encoding process in encoder $E$, the corresponding representations are obtained after encoding each sequence, denoted as $e_1^{(1)}, e_1^{(2)}, \cdots, e_n^{(m)}$. As in the sequence-to-sequence model, an autoregressive decoder is used to obtain the final representation of each input. When decoding, to obtain the semantic representation of the sentence pair, the special token [CLS] is appended to both the start and end of the input sequence, in the form of [CLS], $u_1, u_2, \cdots, u_m$, [SEP], $p_1^{(1)}, p_1^{(2)}, \cdots, p_1^{(m)}$, [CLS]. Then, the [CLS] token is fed into a fully connected layer to obtain the classification score. Finally, the score of each persona is given into a softmax layer to obtain the normalized probability. Take the persona with the highest score as the appropriate character $P_{adapted}$.

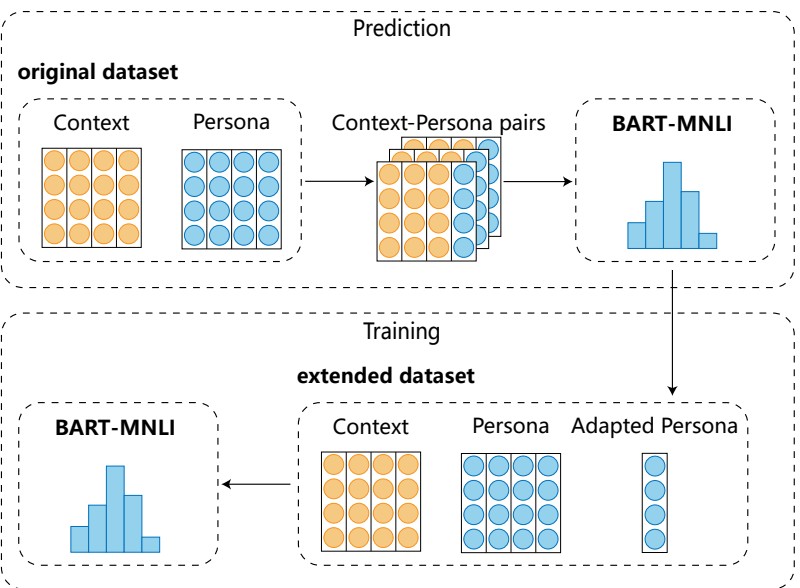

**Figure 4.** The traning and prediction processes of classifier $C_{ps}$. The prediction process is applied twice, firstly to construct the expanded dataset, where the response replaces the context. The second is to give the adapted persona when training our model. The training process uses the expanded dataset to fine-tune the classifier and improve the accuracy of the persona dataset.

*3.5. Decoder $D_1$*

The persona-adapted response generation decoder is an auto-regressive model and is initialized using BERT to ensure the robustness of the response.

For the typical transformer-based dialogue generation decoder, as shown in Figure 3, a cross-attention is applied to incorporate the context information into the hidden vector $O_{merge}$; cross-attention is as follows:

$$r_1^{i+1} = \mathrm{MultiHead}(O_{merge}^{i+1}, H, H) \tag{4}$$

In contrast to the multi-head attention in encoder, which is a self-attentive mechanism that focuses on the relations of words in the hidden vector, cross-attention focuses on the dependence of the hidden vector $r_1$ on the context representation $H$. In training, due to the autoregressive decoder $D_1$, the words after the current prediction should be invisible, so a left-to-right mask is performed in decoder $D_1$, where $r_0$ in decoder $D_1$ is the embedding of the target response.

In addition, to maintain a consistent conversation style for the responses, an auxiliary task is employed in cross-attention, where the adapted persona obtained by classifier $C_{ps}$ prompts the agent to exhibit an adapted speaking style. To implement the auxiliary task in which the hidden vector is revised depending on the adapted persona, an attention routing

mechanism is applied. Attention routing effectively merges the three types of information. The attentional routing is as follows:

$$O^{i+1}_{adapted} = \text{MultiHead}(r^i_1, p_{adapted}, p_{adapted}) \tag{5}$$

$$O^{i+1}_{context} = \text{MultiHead}(r^i_1, E_c, E_c) \tag{6}$$

$$O^{i+1}_{prev} = \text{MultiHead}(r^i_1, r^i_1, r^i_1) \tag{7}$$

$$O^{i+1}_{adapted} = O^{i+1}_{adapted} + O^{i+1}_{context} + O^{i+1}_{prev} \tag{8}$$

Notice that to save computational resources, the multi-head attentions in decoder $D_1$ share parameters. With the same configuration as the encoder $E$, the decoder module also has $N$ layers, and the output $r_1$ of the last layer is taken as the original response, denoted as $R_1$, which is passed to $D_2$ for further processing.

### 3.6. Decoder $D_2$

As in encoder $E$ and decoder $D_1$, decoder $D_2$ is initialized with BERT in order to generate fluent, coherent responses.

In decoder $D_2$, responses are rewritten to merge the original responses from $D_1$ into the hidden vector. Before this, an auxiliary task is performed to revise the inconsistent persona in the original response by a cross-attention to the relation between the hidden vector and the given persona. The procedure is as follows:

$$O^{i+1} = \text{FNN}(\text{MultiHead}(r^i_2, E_p, E_p)) \tag{9}$$

$$r^{i+1}_2 = \text{FNN}(\text{MultiHead}(O^{i+1}, R_1, R_1)) \tag{10}$$

where two multi-head attentions share parameters of the attention matrix. In the auxiliary training, the whole input should be fed to the multi-head attention, and we do not mask the input. After (10), the rewrite response $r_2$ is obtained, which revises the inconsistent personas in it. The output of the last layer of $D_2$, denoted as $R_2$, is the representation of the final response. At last, $R_2$ is fed into a linear output layer, and the agent generates the final response $R$.

### 3.7. Training

In this work, negative log-likelihood (NLL) loss is used for the response generation task, and unlikelihood loss is used for the auxiliary tasks.

#### 3.7.1. Response Generation Task

The negative log-likelihood loss is used in the response generation task to close the generated responses to the target responses. Encoder $E$ and decoder $D_1$ predict the target response according to the personas and context, and the output is the original response $R_1$:

$$\mathcal{L}^{D_1}_{NLL} = -\sum_{i=1}^{|R|} log P_\phi(R^{(i)}|P, C, p_{adapted}, R^{(<i)}) \tag{11}$$

Decoder $D_2$ is trained in the same way to predict the target response according to personas and context:

$$\mathcal{L}^{D_2}_{NLL} = -\sum_{i=1}^{|R|} log P_\gamma(R^{(i)}|P, C, R^{(<i)}) \tag{12}$$

The loss function of this task is $\mathcal{L}_1 = \mathcal{L}^{D_1}_{NLL} + \mathcal{L}^{D_2}_{NLL}$

### 3.7.2. Auxiliary Tasks

In this paper, the adapted persona and consistency inference tasks are both learned from positive and negative samples; therefore, both utilize unlikelihood loss. In the adapted persona task, we use the dataset obtained by the classifier. Moreover, following the practice in contrast learning, we divide the samples into positive samples that are adapted personas, hard negative samples that are other personas of the same user as the positive samples, and soft negative samples that are the personas of other users. Positive samples are from the pairs of context and adapted persona in the corresponding response, donated as $P_{adapted}, R$. Hard negative samples are from the pairs of context and other personas in the corresponding response, and soft negative samples are from the random personas, donated as $P_{other\&rand}$. The ratio of $P_{adapted}, R, P_{other\&rand}$ is 2:1:1. The loss function of persona selection is as follows:

$$S_1^+ = \left\{ (P_{adapted}, R) \right\}, S_1^- = \{ (P_{other\&rand}, R) \} \tag{13}$$

$$\mathcal{L}_{UL}^{D_1^+} = -\sum_{i=1}^{|R|} log P_\phi(R^{(i)}|p_{adapted}, C, R^{(<i)}) \tag{14}$$

$$\mathcal{L}_{UL}^{D_1^-} = -\sum_{i=1}^{|R|} log(1 - P_\phi(R^{(i)}|p_{other\&rand}, C, R^{(<i)})) \tag{15}$$

Consistency inference task using a non-dialogical inference dataset. A sample consists of a premise and a hypothesis, denoted as $P^*, R^*$. The positive samples are from the entailed category, and the negative samples are from the contradicted category:

$$S_1^+ = \{ (P^*, R_{entail}^*) \}, S_1^- = \{ (P^*, R_{contradict}^*) \} \tag{16}$$

$$\mathcal{L}_{UL}^{D_2^+} = -\sum_{i=1}^{|R|} log P_\gamma(R_{entail}^{*(i)}|P^*, R_{entail}^{*(<i)}) \tag{17}$$

$$\mathcal{L}_{UL}^{D_2^-} = -\sum_{i=1}^{|R|} log(1 - P_\gamma(R_{contradict}^{*(i)}|P^*, R_{contradict}^{*(<i)})) \tag{18}$$

The loss function for this task is $\mathcal{L}_2 = \beta\mathcal{L}_{UL}^{D_1^+} + (1-\beta)\mathcal{L}_{UL}^{D_1^-} + \theta\mathcal{L}_{UL}^{D_2^+} + (1-\theta)\mathcal{L}_{UL}^{D_2^-}$

### 3.7.3. Classifier Task

The classifier uses negative log-likelihood loss to infer that the predicted persona is likely to be the target persona, and the classifier employs context to predict adapted personas:

$$\mathcal{L}_{C_{ps}} = -\sum_{i=1}^{n+1} log P_\varphi(y_{label}|p_i, C) \tag{19}$$

where $y_{label}$ is the label indicating whether $p_i$ is the adapted persona for C, with 0 for false and 1 for true. A separate NLI-based model trains the persona selection task in the experiments. Therefore, the total loss of the model is the sum of $\mathcal{L}_1, \mathcal{L}_2$.

## 4. Experiments

### 4.1. Datasets

To evaluate the effectiveness of our model on various-persona-distribution datasets, we performed experiments on a persona-dense dialogue corpus **PersonaChat** and a persona-sparse dialogue corpus **PersonalDialog**, which are both publicly available datasets, as shown in Table 1.

Specifically, PersonaChat is a crowd-sourced dataset that collects multi-turn conversations between pairs of crowd-sourced workers with given personas, hence it is persona-dense. ConvAI2 PersonaChat is a version of PersonaChat on the ConvAI2 competition [29],

and we used it as the dataset in our experiments. PersonalDialog is a large-scale dataset that collects user posts from the Chinese social media Weibo. As mentioned in the introduction, in daily dialogue, responses are persona-sparse, and most of the responses in the dataset are irrelevant to the persona profile. Among the test sets given by PersonalDialog are a random test set which is distributed similarly to the training set, and a biased test set which samples the persona-relevant responses. In the experiments, we just adopted the random test set. The statistical results for both datasets employed in the experiment are shown in Table 2.

**Table 1.** The examples of PersonaChat and PersonalDialog.

|  | **PersonaChat** |
| --- | --- |
| Context | A: hi, how are you doing today?<br>B: i am spending time with my 4 sisters. what are you up to.<br>A: wow, four sisters. just watching game of thrones. |
| Persona | my mom is my best friend.<br>i love iced tea.<br>i have four sisters. |
| Response | that is a good show. i watch that while drinking iced tea |
|  | **PersonalDialog** |
| Context | A: You're going to the gym?<br>B: I exercise for half an hour every day.<br>A: I run over 200 km a month. |
| Profile | interset tags: iPhone;Apple.<br>location: Singapore<br>gender: male |
| Response | Then I'm no match for you |

**Table 2.** The statistical results of persona-dense and persona-sparse open datasets.

| Dataset | Statistics | Train | Valid | Test | Total |
| --- | --- | --- | --- | --- | --- |
| PersonaChat | Dialogues | 16,090 | 1788 | 1000 | 18,878 |
|  | Avg utterances [1] | 14.7 | 14.7 | 15.6 | 14.8 |
|  | Avg personas [2] | 4.5 | 4.5 | 4.5 | 4.5 |
| PersonalDialog | Dialogues | 5,438,165 | 10,000 | 10,000 | 5,458,165 |
|  | Avg utterances | 2.6 | 6.0 | 6.0 | 2.7 |
|  | Avg personas | 2.0 | 2.0 | 2.0 | 2.0 |

[1] Average utterances in each dialogue. [2] Average personas in each dialogue.

However, the raw corpus set contains only the context and the persona of the speaker, without the adapted persona corresponding to the utterances used in the training of our proposed persona selection classifier. As a solution, we use an NLI model to predict the persona implied in the responses, initialized by BART-MNLI [17]. As shown in Figure 4, context and personas are fed as text and classification labels to obtain scores for each label, and the persona with the top score is adopted as the implied persona. In addition, due to the existence of persona-irrelevant utterances, we append the "None" persona to the personas so that the model can determine whether the utterance is persona-irrelevant. We randomly sampled 200 classification results for validation, and the classification accuracy achieved 88.5%. The implied persona in the response depends on the speaking style in the context, i.e., the adapted persona. This extended dialogue dataset was employed for the classifier training.

Additionally, for the consistency inference task, two non-dialogue inference datasets, MNLI [30] and the corresponding Chinese dataset CMNLI [31], were applied.

*4.2. Baselines*

In our experiments, we compared our proposed model with the following strong baselines:

4.2.1. Universal Language Model

**DialogWAE** [32] proposes a conditional Wasserstein auto-encoder, and models the distribution of data with a GAN for dialogue modeling. **GPT2** [33] is a transformer-based language model that reached state-of-the-art performance on the various tasks in 2019. **OPT** [34] is a large-scale transformer-based model and recently open-sourced, with performance similar to that of GPT3, with the full model reaching 175B parameters, and we adopted the released version with 350M parameters.

4.2.2. Persona-Based Model

**Persona-CVAE (PerCVAE)** [4] is a conditional variational auto-encoder (CVAE)-based model, which employs a memory-augmented mechanism to fuse persona information into the context. **Transformer** [29] is the baseline of the pre-trained model, and the architecture achieved state-of-the-art performance in the ConvAI2 competition. **Attention routing (AR)** [18] incorporates the target persona information and dialogue history via the proposed AR mechanism into the hidden state and balances the contribution. **BERT Over BERT (BoB)** splits the persona-relevant dialogue generation into two subtasks and designs the dual-decoder structure. To maintain persona consistency capability in our model, natural language inference data are used to train the decoder $D_2$.

*4.3. Experimental Settings*

Firstly, context, personas, and responses are tokenized with a shared vocabulary, which is the same as that of BERT, with a size of 30,000. Then, the tokenized text by the WordPiece method is fed into the embedding layer with a dimension of 768. We set the length of the context to 10 sentences and the maximum length of the tokenized text to 64 and filled the blanks that had not reached the maximum length with [pad]. For the multi-head attention in the encoder and decoder, the number of heads was set to 8 unless otherwise specified. For training, the dimension of the hidden layer is 768, and the hidden layer is 12, the batch size was set to 16, and the optimizer was Adam, where we used the warmup strategy up to 6000 steps with a learning rate of 3e-6 and later with a learning rate of 7e-6. We selected the trained model that performed well on the validation set for testing.

For comparison experiments with the baseline model, we took the publicly available open-source code and tested them on the two datasets we used, with as little change as possible. In our experiments, we ensured that the baseline model used achieved the performance shown in their paper.

*4.4. Evaluation Metrics*

To show the performance of different models in terms of language understanding, persona consistency, and persona adaptation, we carried out automatic and manual evaluations.

4.4.1. Automatic Evaluation

To estimate language understanding of models, we employed the following metrics: **perplexity (ppl.)** indicates how similar the responses are to the test data. Lower perplexity means a better language model. **Distinct (Dist.)** [35] is used to measure the diversity of the text, specifically the proportion of unique n-grams in the responses, where n is 1, 2, and the higher the distinct, the better the diversity of the responses generated by the model.

For persona consistency, we employed the following metrics: **Delta perplexity (ΔP)** [36] presents the consistency of the generated responses. Specific **p.Ent**, **p.Ctd**, p.Ent, and p.Ctd are the perplexity of entailed and contradicted sentences in the inference dataset.

Higher delta perplexity means the model is better in persona consistency. **Consistency score (C.Score)** [37] is a metric indicating the persona-consistency of the response, as follows:

$$\text{NLI}(r, p_j) = \begin{cases} 1 & \text{if } r \text{ entails } p_j \\ 0 & \text{if } r \text{ is irrelevant to } p_j \\ -1 & \text{if } r \text{ contradicts } p_j \end{cases} \tag{20}$$

$$C(r) = - \sum_{j}^{m} \text{NLI}(r, p_j) \tag{21}$$

The responses are classified by an inference model, and for fairness, we adopted the RoBERTa [38] model. Without fine-tuning, the model achieves an accuracy of 87.6% on the test dataset.

To show the ability of the model to select adapted characters, we designed a metric, **persona accuracy (Per.Acc)**, defined as follows:

$$\text{Per.Acc}(R, P) = \frac{count(r | p_{adapted} = p_r)}{count(R)} \tag{22}$$

First, the adapted persona is predicted by an inference model, and for fairness, we used a RoBERTa model that is trained in the same way as classifier $C_{ps}$. The persona implied in the response $p_r$ is predicted by the model in C.Score. The ratio of responses with $p_{adapted}$ as $p_r$ in the total responses is Per.Acc.

### 4.4.2. Human Evaluation

We employed five testers in our experiment, independent of the model designers, who are fluent in the language of the annotated data. We randomly sampled 100 test cases containing personas, contexts, and responses, and the testers needed to evaluate them on utterance fluency, dialogue diversity, and persona consistency. Testers were instructed to rate the responses on a scale from 1 to 5, where 1 means very bad, 3 means average, and 5 means very good. For example, if the context is "How is your job?", the persona is "I am a lawyer." and the response is "I am a doctor and I see many patients every day.", the tester should give a score of 5 for fluency, 5 for diversity, and 1 for consistency.

## 5. Result

Tables 3 and 4 show the performance of the baselines and our proposed models for PersonaChat and PersonalDialog, and the results in bold mean the best performance. In addition, we performed ablation experiments for each module as shown in Tables 5 and 6. In Table 7, we also explore the persona distribution in different models. Finally, with the case study in Table 7, we illustrate the superiority of our model over baselines.

**Table 3.** Experiment results of PersonaChat dataset.

| Model | ppl. | Dist.1 | Dist.2 | p.Ent | p.Ctd | ΔP | C.Score | Per.Acc | Fluency | Diversity | Consistency |
|---|---|---|---|---|---|---|---|---|---|---|---|
| DialogWAE [32] | 37.4 | 3.24 | 14.96 | 35.8 | 41.7 | 5.9 | 1.74 | 2.21 | 3.12 | 3.03 | 3.07 |
| GPT2 [33] | 12.7 | 7.68 | 28.42 | 11.3 | 20.6 | 9.3 | 14.73 | 8.76 | 3.93 | 3.47 | 3.34 |
| OPT [34] | 6.6 | 8.95 | 31.83 | 6.4 | 48.5 | 42.1 | 16.47 | 9.24 | 4.16 | 3.63 | 3.78 |
| PerCVAE [4] | 41.5 | 2.76 | 12.59 | 38.5 | 45.7 | 7.2 | 7.92 | 6.94 | 2.91 | 2.85 | 3.26 |
| Transformer [29] | 25.3 | 4.39 | 19.87 | 22.6 | 35.8 | 13.2 | 2.83 | 3.02 | 3.36 | 3.14 | 2.69 |
| AR [18] | - | - | - | - | - | - | - | - | - | - | - |
| BoB [9] | 7.0 | 8.60 | 27.94 | 6.7 | 79.3 | 72.6 | 18.64 | 9.73 | 3.87 | 3.61 | 3.81 |
| Ours | **4.3** | **9.31** | **33.18** | **4.1** | **92.5** | **88.4** | **18.91** | **14.57** | **4.24** | **3.72** | **4.17** |

**Table 4.** Experiment results of PersonalDialog dataset.

| Model | ppl. | Dist.1 | Dist.2 | p.Ent | p.Ctd | ΔP | C.Score | Per.Acc | Fluency | Diversity | Consistency |
|---|---|---|---|---|---|---|---|---|---|---|---|
| DialogWAE [32] | 52.4 | 2.52 | 8.92 | 50.2 | 54.9 | 4.7 | -1.95 | 0.16 | 2.57 | 2.26 | 2.07 |
| GPT2 [33] | 24.0 | 4.67 | 16.59 | 23.5 | 49.8 | 26.3 | 2.47 | 1.67 | 2.98 | 2.75 | 3.04 |
| OPT [34] | 16.8 | 5.82 | 18.52 | 15.4 | 67.3 | 52.3 | 3.83 | 1.93 | 3.44 | 2.93 | 3.42 |
| PerCVAE [4] | 58.5 | 2.38 | 7.53 | 52.3 | 65.2 | 12.9 | 0.81 | 0.35 | 2.21 | 2.07 | 2.35 |
| Transformer [29] | 39.1 | 2.82 | 11.58 | 38.4 | 44.1 | 5.7 | 0.54 | 0.28 | 2.63 | 2.36 | 2.39 |
| AR [18] | 32.6 | 4.45 | 15.96 | 27.7 | 56.2 | 28.5 | 1.32 | 0.91 | 2.65 | 2.58 | 2.78 |
| BoB [9] | 17.4 | 5.62 | 18.31 | 15.2 | 72.5 | 57.3 | 3.51 | 1.27 | 3.04 | 2.85 | 3.34 |
| Ours | **12.9** | **6.17** | **23.70** | **10.8** | **93.6** | **82.8** | **4.19** | **2.84** | **3.59** | **3.21** | **3.62** |

**Table 5.** Ablation study results of PersonaChat dataset.

| Model | ppl. | Dist.1 | Dist.2 | p.Ent | p.Ctd | ΔP | C.Score | Per.Acc | Fluency | Diversity | Consistency |
|---|---|---|---|---|---|---|---|---|---|---|---|
| Ours | 4.3 | 9.31 | 33.18 | 4.1 | 92.5 | 88.4 | 18.91 | 14.57 | 4.24 | 3.72 | 4.17 |
| w/o $UL_{D_1}$ | 5.6 | 8.95 | 32.62 | 4.9 | 78.2 | 73.3 | 16.75 | 8.35 | 4.01 | 3.69 | 4.06 |
| w/o $UL_{D_2}$ | 6.1 | 8.71 | 31.73 | 5.9 | 22.8 | 16.9 | 3.32 | 12.94 | 3.95 | 3.58 | 3.31 |
| w/o $UL_{D_1}$&$UL_{D_2}$ | 6.9 | 8.69 | 29.97 | 6.8 | 21.1 | 14.3 | 3.38 | 7.92 | 3.76 | 3.39 | 3.24 |
| w/o $D_2$ | 20.8 | 3.73 | 17.63 | 19.4 | 21.5 | 2.1 | 2.84 | 10.27 | 3.87 | 3.49 | 3.29 |
| E+D1 | 24.2 | 3.52 | 15.88 | 23.3 | 26.1 | 2.8 | 2.79 | 3.72 | 3.44 | 3.25 | 3.17 |

**Table 6.** Ablation study results of PersonalDialog dataset.

| Model | ppl. | Dist.1 | Dist.2 | p.Ent | p.Ctd | ΔP | C.Score | Per.Acc | Fluency | Diversity | Consistency |
|---|---|---|---|---|---|---|---|---|---|---|---|
| Ours | 12.9 | 6.17 | 23.70 | 10.8 | 93.6 | 82.8 | 4.19 | 3.84 | 3.59 | 3.21 | 3.62 |
| w/o $UL_{D_1}$ | 13.8 | 5.75 | 21.41 | 11.3 | 78.6 | 67.3 | 3.93 | 1.73 | 3.38 | 3.19 | 3.26 |
| w/o $UL_{D_2}$ | 14.7 | 5.61 | 20.37 | 12.9 | 26.3 | 13.4 | 2.17 | 3.04 | 3.47 | 3.03 | 3.31 |
| w/o $UL_{D_1}$&$UL_{D_2}$ | 15.3 | 5.64 | 19.70 | 13.0 | 24.1 | 11.1 | 2.01 | 1.69 | 3.26 | 3.14 | 3.08 |
| w/o $D_2$ | 38.9 | 2.03 | 9.74 | 35.8 | 41.7 | 5.9 | 1.48 | 2.95 | 2.84 | 2.93 | 2.53 |
| E+D1 | 43.4 | 1.66 | 8.31 | 41.2 | 43.4 | 2.2 | 1.39 | 1.26 | 2.73 | 2.95 | 2.04 |

### 5.1. Model Performance

As shown in Tables 3 and 4, our proposed model obtained better performance on all automatic and manual metrics than other models on both datasets, indicating that our model effectively utilizes persona information to generate persona-adapted and persona-consistent high-quality responses. Specifically, on the conversation generation task, compared to the large-scale pre-trained model OPT, our models achieve better fluency and diversity. This result is attributed to the framework of the dual decoder and the attention-routing mechanism, because BoB, which uses the dual decoder, and AR, which uses attentional routing, do not perform better than OPT. Unlike BoB, AR uses a single-sentence query as input. Another reason is that our model learns from multi-turn context and can obtain sentence-level hidden vectors to improve fluency. In addition, we observe that DialogWAE and PerCVAE have an obvious gap with other baselines. This demonstrates that the attention mechanism of the large-scale pre-trained model has a substantial enhancement for the conversation generation task. Benefiting from the transformer structure, our model has a massive advantage in language understanding over earlier models. Comparing the performance of PerCVAE and DialogWAE, which are similar in structure, we observe two phenomena: 1. The responses of PerCVAE contain more persona information, which demonstrates the effectiveness of the fused persona information process. 2. PerCVAE has no significant improvement in persona consistency understanding, demonstrating the necessity to constrain the process of fusing persona information. Moreover, our model and BoB achieve the top two on the consistency metric, which shows that the process of fusing persona information should be constrained. Since our proposed persona selection

module learns the persona distribution on different datasets, our model achieves the best performance on speaking style consistency.

*5.2. Further Analysis*

5.2.1. Ablation Study

As stated above, our model performs well, and we analyzed it in four aspects in the model: (1) adapted persona, (2) consistency inference, (3) dual decoder, and (4) multi-turn context. We design a ablation experiment, as in Tables 5 and 6, as follows: **w/o UL$_{\mathbf{D_1}}$** the model removes the persona selection auxiliary task; **w/o UL$_{\mathbf{D_2}}$** the model removes the consistency inference auxiliary task; **w/o UL$_{\mathbf{D_1}}$&UL$_{\mathbf{D_2}}$** the model removes both auxiliary tasks; **w/o D$_{\mathbf{2}}$** the model removes the decoder $D_2$, and since consistency inference is acted on $D_2$, it is removed as well; **w/E + D1** the model removes the decoder $D_2$ and both auxiliary tasks, and the model degenerates to a transformer model.

**The effect of adapted persona**: Adapted personas in our model are predicted by the classifier $C_{ps}$ from the given personas and are consistent with the speaking style implied in the context. The adapted persona representations are fused into the original response $R_1$ through attention routing and trained with a persona-selection auxiliary task. As shown in Tables 5 and 6, comparing the full model with **w/o UL$_{\mathbf{D_1}}$,w/o UL$_{\mathbf{D_2}}$**, and **w/ E+D1**, we can see that the incorporation of adapted persona has a significant improvement on Per.Acc in both datasets, especially in PersonaChat. The explanation is that more responses are persona-relevant in the persona-dense dataset, and the upper limit of the persona distribution is higher.

**The effect of consistency inference**: The consistency inference task works as an auxiliary task of decoder $D_2$ to reduce inconsistent persona information in the rewritten responses. Compared with the full model, the **w/o UL$_{\mathbf{D_2}}$** experiments have a significant decrease in ΔP and manually evaluated consistency in Tables 5 and 6. We can assume that the consistency inference task is key to the semantic consistency of personas.

**The effect of dual decoder**: The dialogue generation process in the model consists of two individual decoders. There are two doubts about the dual decoder: (1) Does rewriting the decoder have an effect? (2) Is the improvement in the model just the result of a larger-scale model? For question (1), we compare **w/o UL$_{\mathbf{D_2}}$** with **w/o D$_{\mathbf{2}}$** and find that decoder $D_2$ shows a vast improvement in perplexity, distinct 1/2, and manually evaluated fluency and diversity, and we determine that the larger model has a better performance on language understanding. For question (2), the larger model scale is not the only reason for the good performance of our model. Firstly, we compare our models, as shown in Tables 5 and 6, and find that the full model has a significant improvement in all aspects than **w/o UL$_{\mathbf{D_1}}$ and UL$_{\mathbf{D_2}}$**, which confirms that the two auxiliary tasks we proposed contribute to all metrics. In addition, compared with a model with a dual-decoder structure, BoB, persona selection task, and attention routing significantly contribute to our model in Tables 3 and 4.

**The effect of multi-turn context**: Since persona selection in our model refers to context, we chose multi-turn context for our inputs instead of single-turn queries. To study the effect of multi-turn contexts on our model, we conducted a **w/E + D1** experiment, where the model differs from the transformer model only in the input. Comparing the results of the two models, we find that the results of **w/E + D1** outperform transformer on all metrics in PersonaChat, in Table 5. However, the results are opposite on the PersonalDialog in Table 6. We guess that persona information hinders language understanding on sparse datasets and affects the quality of the generated responses. This is why we use the persona selection task to focus on persona distribution.

5.2.2. Study of Persona Distribution

To study the performance of the model on speaking style consistency, we show the persona distribution of our model and baselines on sparse and dense datasets, as shown in Table 7. Comparing the two datasets, we find a huge gap in the real distributions, which

is a great challenge for the generalization performance in the model. We can see that the persona distribution of the non-persona-based model is stable and without apparent changes in both datasets. While the other persona-based models have minor changes, the gap with the persona distribution in datasets is large. In contrast, our model has the smallest gap with the true distribution.

**Table 7.** Persona distribution of result on PersonaChat and PersonalDialog datasets.

|  | **PersonaChat** | **PersonalDialog** |
|---|---|---|
| Real distribution | 46.9% | 1.2% |
| DialogWAE [32] | 7.7% | 6.4% |
| GPT2 [33] | 15.0% | 13.7% |
| OPT [34] | 24.3% | 17.6% |
| PerCVAE [4] | 13.4% | 10.1% |
| Transformer [29] | 11.5% | 10.4% |
| AR [18] | - | 9.4% |
| BoB [9] | 20.0% | 8.6% |
| Ours | **31.7%** | **3.2%** |

5.2.3. Case Study

Besides the quantitative analysis of the model, we also carried out a case study to illustrate the superiority of our model compared to the optimal persona-based and non-persona-based baselines, as shown in Table 8. We can see that the OPT model generates responses that fuse context and persona information, leading to high-quality sentences. However, it contradicts the given persona "i'm 19". The BoB model focuses on persona information without selecting the most adapted persona, resulting in responses irrelevant to the context. Our model selects the adapted persona and responses to the context "where do you want to go?" and generates the best response.

**Table 8.** A sample case generated by baselines and our model on PersonaChat.

| | |
|---|---|
| Persona | i just want to have fun with my friends.<br>i'm 19 and i cannot wait to move out 19 my parents home.<br>i do not drink or do drugs or anything.<br>i work took i have a part time job at burger king. |
| Adapted Persona | i just want to have fun with my friends. |
| Context | A: hi there ! sitting on the porch with my kitty .<br>B: i'm wishing i could go out with my friends but mom and dad won't let me<br>A: awww ! i'm sorry ! where do you want to go? |
| OPT [34]<br>BoB [9]<br>Our<br>Gold Response | It's not fair that I can't move out with my friends just because I'm only 20.<br>Hey there! I'm doing okay, just working a part-time job at Burger King right now.<br>anywhere with good friends. i want to go to a new place.<br>i want to move out so i can have fun with my friends whenever i want. |

**6. Conclusions**

In this study, we propose a novel persona-based dialogue model to address persona stylistic and semantic consistency in multi-turn dialogue. We selected the adapted persona by an NLI-based classifier according to the context, and aligned the persona's distribution in the responses by an attention-routing mechanism to achieve consistency of speaking style in the decoder $D_1$. We trained the rewriting decoder D2 through the NLI dataset and realized persona semantic consistency. The experiments on two datasets with different distributions of persona conversations, i.e., persona-dense PersonaChat, and persona-sparse PersonalDialog, show that our proposed method has a distinct improvement over the advanced persona model. We also conducted ablation experiments to study the effect of the persona selection mechanism, proving that persona selection is effective for matching

the persona distribution in the dataset. In future work, we will try to extract persona information from conversation interactions and enhance semantic representations by graph structures to represent persona information.

**Author Contributions:** Conceptualization, S.Z. and T.M.; methodology, S.Z. and H.R.; software, S.Z.; validation, S.Z.; formal analysis, S.Z.; supervision, H.R. and N.A.-N.; investigation, S.Z. and N.A.-N.; resources, S.Z.; data curation, S.Z.; writing—original draft preparation, S.Z.; writing—review and editing, S.Z. and T.M.; project administration, T.M. and H.R. All authors have read and agreed to the published version of the manuscript.

**Funding:** This research was funded by National Key Research and Development Program of China (2021YFE0104400), the National Natural Science Foundation of China (NO. 62102187).

**Institutional Review Board Statement:** Not applicable.

**Informed Consent Statement:** Not applicable.

**Data Availability Statement:** Not applicable.

**Conflicts of Interest:** The authors declare no conflict of interest.

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
