# Peer review of "A Personalized Multi-Turn Generation-Based Chatbot with Various-Persona-Distribution Data"

_applsci, doi:10.3390/app13053122_

Round 1
Reviewer 1 Report
Most references are from 2019 to 2020, lacking some latest technical descriptions The baseline algorithm in the comparison experiment is not new, and the reference of the comparison algorithm should be added in the table
The experiment is sufficient, but the analysis is mostly surface data analysis, lacking some deep theoretical analysis
In the part of algorithm description and experiment, the method of combining graphics and text is easier for readers to understand.
Author Response
Dear editor and reviewer,
On behalf of my co-authors, we are very grateful for giving us an opportunity to revise our manuscript, and we also appreciate reviewers very much for their positive and constructive comments and suggestions on our manuscript entitled “A personalized multi-turn generation-based chatbot with various-persona-distribution data” (Manuscript ID: applsci-2173976).
We revised the manuscript according to these comments and suggestions. In general, we have tried our best to revise our manuscript and provide the point-by-point responses. All changes were marked in BLUE using the “Track Changes” function in the revised manuscript.
Point 1: Most references are from 2019 to 2020, lacking some latest technical descriptions The baseline algorithm in the comparison experiment is not new, and the reference of the comparison algorithm should be added in the table.
Response 1: Thank you for your comments, the work we have revised is as follows:
First, in the Related Work, we select some representative work since 2019, due to the fact that research on persona-based conversations has sprung up since the release of the PersonaChat dataset in 2018. With your valuable comment, we have added the latest research to illustrate the trends of existing research on persona-based models.
We have added four recent references (2021-2023) to the section 2. Specifically, in Line 114-121 and Line 137-138.
Second, we select a latest persona-based model, BoB, and a latest pre-trained language model, OPT, as two strong baselines and our models achieve better results than strong baselines.
We update the results table 3,4,7,8 and our model is the best one.
Point 2: The experiment is sufficient, but the analysis is mostly surface data analysis, lacking some deep theoretical analysis.
Response 2: Thank you for your reminder. We have revised the analysis in Section 5.1. We have added the theoretical analysis in Line 371-376, 380-386, 390-394 to illustrate the efficiency of the module. Overall, the modules of the model played an essential role in the experiment.
Point 3: In the part of algorithm description and experiment, the method of combining graphics and text is easier for readers to understand.
Response 3: Thank you for your comments. We have added Figure 4, Table 2, and the corresponding text to better describe classifier Cps and the datasets. For the other modules we have also combined the figures to describe them.
Once again, thank you very much for your comments and suggestions. And we hope that the revised manuscript can be accepted by Applied Sciences.
Yours sincerely.
Shihao Zhu, Tinghuai Ma*, Huan Rong, Najla Al-Nabhan
Corresponding author: Tinghuai Ma * (thma@nuist.edu.cn)

Reviewer 2 Report
This study presents the development of a complete dialogue generation system that leverages persona information to increase semantic and persona consistency of responses in multi-turn dialogues. It first selects a persona using an NLI classifier and then rewrites the answer considering conversation consistency. The presented model architecture consists of several parts in sequential order, all of which significantly influence the final model performance as the ablation study showed. In addition, a thorough evaluation was conducted as several persona-based and language models have been compared on two different datasets. Overall the paper is well-written and well-structured.State of the art is discussed adequately. The evaluation performed is thorough, also including an ablation study, which is good.
A few minor enhancements are the following:
Line 27-28 please explain the difference between the 3 categories with examples
Line 47 what exactly is “persona-relevant”? How do we label a response are persona relevant or irrelevant?
Figure1b spelling, fan not fans.
Section 4.1 provide examples from the two datasets for a better understanding.
Section 4.4.2 give examples of responses with their ratings.
spelling: BERT not bert
Author Response
Response to Reviewer 1 Comments
Dear editor and reviewer,
On behalf of my co-authors, we are very grateful for giving us an opportunity to revise our manuscript, and we also appreciate reviewers very much for their positive and constructive comments and suggestions on our manuscript entitled “A personalized multi-turn generation-based chatbot with various-persona-distribution data” (Manuscript ID: applsci-2173976).
We revised the manuscript according to these comments and suggestions. In general, we have tried our best to revise our manuscript and provide the point-by-point responses. All changes were marked in BLUE using the “Track Changes” function in the revised manuscript.
Point 1: Line 27-28 please explain the difference between the 3 categories with examples.
Response 1: Thanks to your suggestion, we have added Figure 1 and the corresponding text in Line 28-33, using three examples to illustrate the purpose and differences amony the three types of consistencies.
Point 2: Line 47 what exactly is “persona-relevant”? How do we label a response are persona relevant or irrelevant?
Response 2: Thanks to your suggestion, we have revised lines 40-42 to explicitly define "persona-relevant" as follows:
we refer to these sentences containing persona information as persona-relevant responses.
Where persona-relevant does not mean that a response contains the correct persona information, for example, earlier models did not incorporate persona information with appropriate constraints, and that is why we need to consider persona consistency.
In our work, labeling responses is a part of building the dataset for classifier Cps. We illustrate this process in Section 3.4 and lines 280-286 of Section 4.1. Briefly, we use the Bart-MNLI model to score the pair {response, personas} to get the most relevant persona, which is a multiclassification task. To determine whether a response is persona-relevant, we add a "None" persona to the personas, which is a simple and really effective method. When the response is persona-irrelevant, "None" scores significantly higher than the other personas, and in line 287 we illustrate that this automatic method has an accuracy of 88.5% by manual evaluation.
Point 3: Figure1b spelling, fan not fans.
Response 3: Thank you for your careful attention, we have corrected the spelling error in Figure 2b.
Point 4: Section 4.1 provide examples from the two datasets for a better understanding.
Response 4: Thanks to your suggestion, we have added Table 2 in section 4.1 to show examples from the PersonaChat and PersonalDialog datasets.
This is more useful to show the differences, as follows:
- the difference between persona and profile.
- the majority of PersonaChat responses are persona-relevant and the majority of PersonaDialog are persona-irrelevant, i.e., different persona distributions.
Point 5: Section 4.4.2 give examples of responses with their ratings.
Response 5: Thank you for your suggestion, we have added a sample in Line 357-359 of section 4.4.2 to illustrate how the testers scored the responses, the additions are as follows:
For example, the context is "How is your job?", the persona is "I am a lawyer." and the response is "I am a doctor and I see many patients every day.", the tester should give a score of 5 for fluency, 5 for diversity and 1 for consistency.
Point 6: spelling: BERT not bert
Response 6: Thank you for pointing out the spelling error, we have corrected the spelling to BERT in full paper.
Once again, thank you very much for your comments and suggestions. And we hope that the revised manuscript can be accepted by Applied Sciences.
Yours sincerely.
Shihao Zhu, Tinghuai Ma*, Huan Rong, Najla Al-Nabhan
Corresponding author: Tinghuai Ma * (thma@nuist.edu.cn)

Round 2
Reviewer 1 Report
nothing